# Use of Chicken Feather Peptone and Sugar Beet Molasses as Low Cost Substrates for Xanthan Production by *Xanthomonas campestris* MO-03

**Murat Ozdal \***  **and Esabi Başaran Kurbanoglu**

Department of Biology, Faculty of Science, Ataturk University, 25240 Erzurum, Turkey; ebasaran@atauni.edu.tr
\* Correspondence: murat.ozdal@yahoo.com; Tel.: +90-442-231-1648

**Abstract:** Xanthan gum is one of the polysaccharides most commonly used in a broad range of industries (food, cosmetics, pharmaceutical, etc.). Agro-industrial by-products are being explored as alternative low-cost nutrients to produce xanthan gum by *Xanthomonas campestris*. In this study, for the production of xanthan gum, sugar beet molasses and chicken feather peptone (CFP) were used as carbon and nitrogen sources, respectively. *X. campestris* produced the highest level of xanthan gum (20.5 g/L) at 60 h of cultivation using sugar beet molasses (40 g/L total sugar) supplemented with CFP (4 g/L) at pH 7, 200 rpm, and 30 °C. The pyruvic acid content of the xanthan gums increased with increasing CFP concentration. Compared with commercial organic nitrogen sources (tryptone, bacto peptone, and yeast extract), the highest production of xanthan gum was obtained with CFP. Moreover, among the tested peptones, the highest pyruvic acid (3.2%, *w/w*) content was obtained from CFP. The usage of sugar beet molasses and CFP as substrates in industries would enable a cost-efficient commercial production. These results suggest that sugar beet molasses and CFP can be used as available low-cost substrates for xanthan gum production by *X. campestris*.

**Keywords:** xanthan gum; *Xanthomonas campestris*; chicken feather peptone; molasses

## 1. Introduction

Many bacteria and fungi are capable of producing polysaccharides [1,2]. One of these polysaccharides is xanthan gum. *Xanthomonas campestris*, one of the phytopathogenic Gram-negative bacteria can convert glucose, sucrose, glycerol, or other organic substrates into xanthan gum. Worldwide consumption of xanthan is estimated at around 80,000 metric tons and the worldwide market of xanthan gum is estimated to be 400 million dollars annually [3]. Owing to a growing worldwide market for polysaccharides by microbial production, xanthan gum has great potential with regard to applications and strategies for production. Because of its industrial applications, it is attracting increasing interest and wide use in a broad range of industries, such as in food, cosmetics, pharmaceutical, paper, paint, textile, and oil [2–5]. Use of xanthan gum in industry applications is promising with large economic potential. Modified xanthan gums also have great potential in tissue engineering and biodegradable edible film applications [6,7].

The production of xanthan gum by microbial fermentation has been shown to be influenced by numerous environmental factors including: Dissolved oxygen level, media composition, temperature, pH and incubation rate, and time [2,8–10]. Using a low-cost substrate for the production of high value products may help to make the process economical. Carbon and nitrogen sources are very important to consider on the selection of wastes as substrates. The costs of the substrates have an important contribution to the overall xanthan gum production cost, and this can be minimized by using cheaper organic wastes. Renewable substrates, such as glycerol (byproduct of biodiesel

production) [5], kitchen waste [11], cheese whey [12], potato starch [13], lignocellulosic agro-industrial wastes [14], and molasses [15] are cheap, easily available and have been investigated as potential substrates for xanthan gum production.

Sugar beet molasses, a byproduct of beet sugar production, is an inexpensive, available, and more usual source of sucrose. It is also a source of nitrogen, minerals, and vitamins [16]. Because of these properties, sugar beet molasses is widely used as a substrate in fermentation studies. Turkey, being an agricultural country, is producing about $8.3 \times 10^5$ tons of beet molasses annually, which is a cheap substrate for microbial fermentation [17].

Approximately six million tons of feathers are produced annually as a waste material in poultry processing plants [18]. This is creating an environmental problem that is difficult to eliminate. Therefore, both convenient and environmentally-friendly, it is necessary to develop an efficient and profitable process to use these by-products. Feathers constitute up to 10% of total chicken weight. They consist of approximately 90% protein composed of keratin, and thus can be a potential source of proteins and amino acids [19–21]. Considering these characteristics of the feathers, they are cheap and abundant for peptone production. Peptones are the hydrolysis products of nitrogen rich substrates. Recently, the chicken feather peptone has been used as a complex nitrogen source for the production of rhamnolipid [19], citric acid [21], and polyhydroxyalkanoate (bioplastic) [22].

Chicken feather peptone (CFP) has not been extensively investigated for bacterial polysaccharide production, and does not have widespread use for microbial production studies. The objective of this study was to use CFP and sugar beet molasses in a fermentation medium for the production of xanthan by using a local isolate of *X. campestris* MO-03.

## 2. Results and Discussion

### 2.1. Effect of CFP on Xanthan Gum Production

*X. campestris* MO-03 strain had been isolated from the infected plant leaf. This strain could produce 14.5 g/L xanthan gum when cultivated in media with the initial glucose concentration of 40 g/L [23]. According to the literature, the best carbon sources are glucose and sucrose for xanthan gum production [2]. Sugar beet molasses contain most substances (especially carbon source) necessary for the nutrition of microorganisms; it can be supplemented with certain components such as nitrogen, phosphorus, or magnesium depending on the fermentation [21].

Figure 1 shows the effects of different CFP concentrations (0–6 g/L) on the sugar consumption, xanthan gum production, and bacterial growth in various incubation times. These experiments have shown that sugar consumption (Figure 1a), bacterial growth (Figure 1b), and xanthan production (Figure 1c) can be affected to differing extents by CFP concentrations. The xanthan gum concentration was observed to be dependent on the CFP concentration.

As expected, the concentration of residual sugars decreased during the fermentation (Figure 1a) and there was an increase in xanthan gum production. For example, the highest xanthan gum concentration in the control medium CM + 4 g/L CFP was observed at 60 h (Figure 1c). This value was 20.5 g/L, and the sugar consumption was 100%. However, the xanthan gum concentration in the CM for the same incubation time was 9.4 g/L, and sugar consumption for this medium was about 70%.

As seen in Figure 1b, the highest biomass yield (3.35 g/L) for 96 h was obtained from 6 g/L CFP, whereas the lowest biomass concentration (2.34 g/L) was obtained in the CM. Biomass production increased with increasing CFP concentrations (Figure 1b), but the maximum xanthan gum production was achieved for 4 g/L CFP (Figure 1c). These results suggested that the increase in xanthan gum concentration was not dependent on the increase in the growth related. Similar results were reported in the previous studies [23–25].

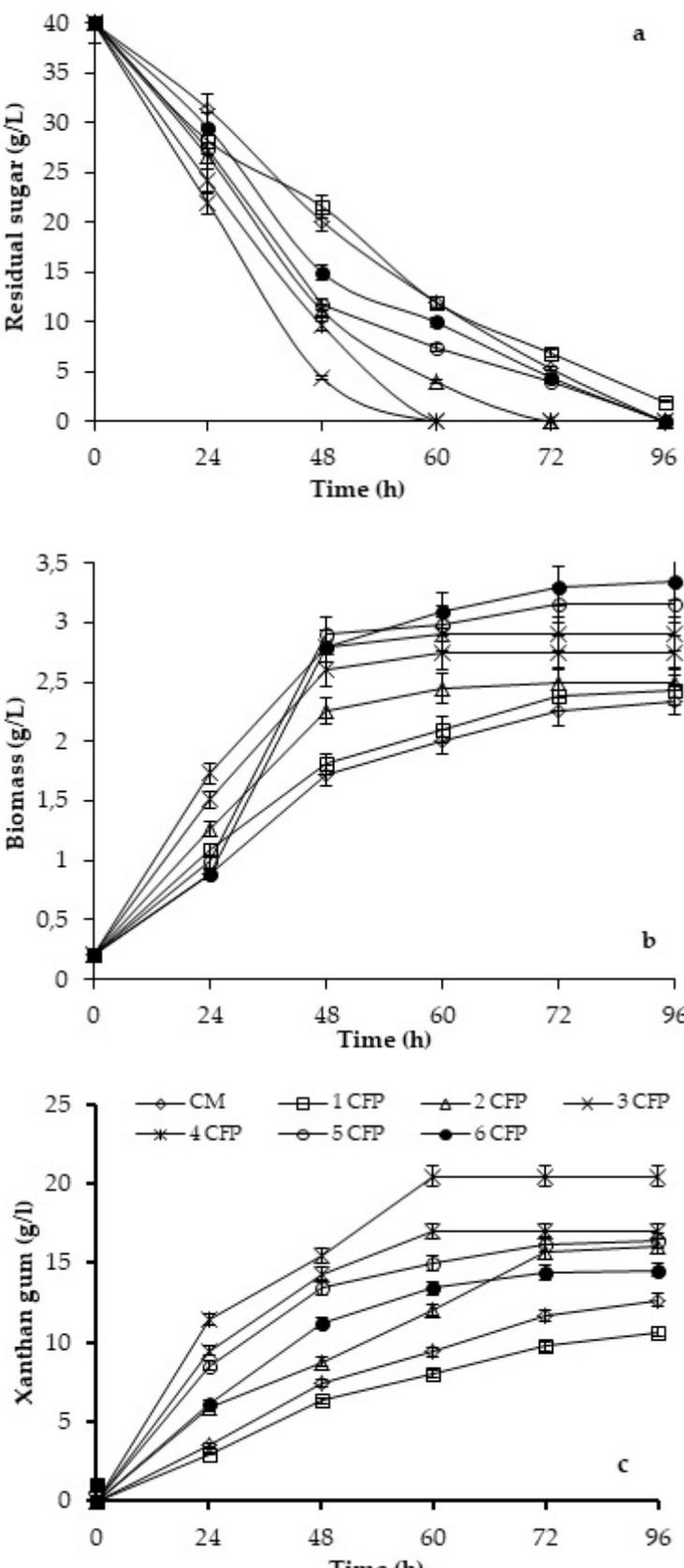

**Figure 1.** Comparison of the effect of chicken feather peptone (CFP) in various concentrations for different incubation times: (**a**) Sugar utilization, (**b**) biomass concentration and (**c**) xanthan gum concentration.

The initial nitrogen concentration and type also affected xanthan production. On account of this, the 4 g/L CFP has a significant effect on the growth of *X. campestris* isolate, as well as the xanthan produced, and the sugar consumed by it after short time. These values for 4 g/L CFP were higher than those for CM. This substrate was termed as molasses chicken feather peptone medium (MCFPM) and its chemical composition was CM + 4 g/L CFP. It was found that applications higher than 4 g/L CFP had a negative effect for xanthan concentration. This negative effect may be a result of high nitrogen source and salt concentration [21,26]. It can be said that the addition of 4 g/L CFP to the CM may provide the proper carbon/nitrogen (C/N) ratio for xanthan gum production. It is well known that higher xanthan gum accumulation occurs at a higher C/N ratio and higher xanthan gum production occurs under nitrogen-limited conditions. For this reason, as the CFP ratio in the media increased, the C/N decreased.

Souw and Demain [27] showed that glutamate was the best nitrogen source for production of xanthan gum. Murad et al. [28] studied the optimization of xanthan production by *X. campestris* using acid hydrolyzed whey as a cheap source of carbohydrate and nitrogen. Researchers [28] also concluded that the cysteine, alanine, and histidine were the best nitrogen sources for the production of xanthan gum. Others have reported that some organic nitrogen sources such as yeast extract [29], ram horn peptone [30], and peptone-tyrptone [28] contribute to the formation of xanthan gum. The results of our previous studies [19,21,23] showed that CFP contains all of these amino acids (alanine, cysteine, glutamate, and proline) at varying concentrations. For the production of xanthan gum, *X. campestris* requires some micro- (e.g., K, Fe, P, Mg, S and Ca salts) and macronutrients (C and N) [31,32]. The increased cell growth and higher xanthan production by CM + 4 g/L CFP in comparison to CM may be due to the presence of amino acids and salts components in the CFP. Moreover, xanthan in the MCFPM (CM + 4 g/L CFP) is 1.55 times higher than that of CM for 60 h. It also reduced the fermentation time from 72 h to 60 h.

The quality of xanthan depends on its pyruvic acid content [33]. The pyruvic acid content of xanthan gum is an important parameter for applications, since it affects the viscosity of the xanthan aqueous solution; low content yields low viscosity, whereas high content promotes gelling [4]. As seen in Figure 2, the addition of CFP significantly improved the quality of xanthan gum ($p < 0.05$); the pyruvate content of xanthan was about 3% (*w/w*), higher than that of the CM (2.1%, *w/w*). Organic nitrogen sources were found to be good for the production of xanthan gum with higher pyruvic acid contents [14,34]. However, the high concentrations of inorganic nitrogen reduced the amount of pyruvic acid of the xanthan gum [2,35]. Moreover, the pyruvate content of the xanthan gum is affected by various fermentation parameters (pH, temperature, media composition, oxygen, and incubation time) [2,9].

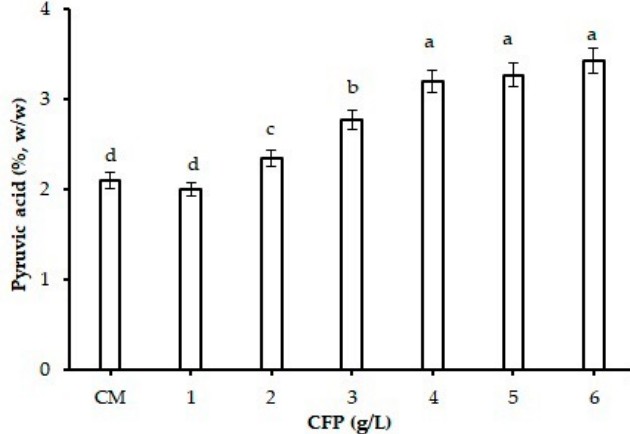

**Figure 2.** Pyruvic acid content of xanthan gums from fermentations carried out at maximum incubation times (60 h of 3–4 CFP; 72 h of 2, 5–6 CFP; 96 h of control and 1 CFP) in molasses fermentation media. Means in the same column followed by the same letter are not significantly different at the $p < 0.05$ level.

### 2.2. Effect of Organic Nitrogen Sources on Production of Xanthan

Four organic nitrogen sources were tested for the production of xanthan gum (Figure 3). The results demonstrated that the maximum biomass yield was obtained with tryptone (TP) (3.6 g/L). Xanthan gum production was the best when CFP (20.5 g/L) was used as a nitrogen source, among all the organic nitrogen sources (TP, BP and YE) tested. The least xanthan gum was obtained in the CM containing YE (16.44 g/L). This positive effect of CFP may be a result of high amino acid and mineral contents [20,23]. The nitrogen contents of CFP, BP, TP and YE are 9.0 [23], 13.8, 10, and 10.9 (g/100 g) [26], respectively. Since CFP is chemically synthesized, the ash content (about 41%) is also high in other organic nitrogen sources (11–15%) [23]. Many researchers reported that the addition of organic nitrogen sources (yeast extract, ram horn hydrolyzate) to the production medium promoted cell growth and xanthan gum production [14,26,36].

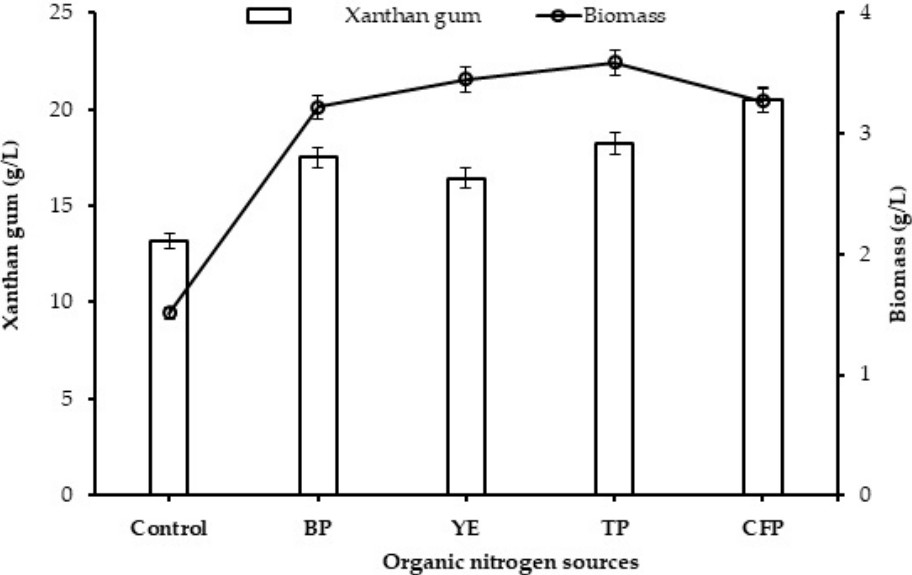

**Figure 3.** Effect of organic nitrogen sources on xanthan gum production and biomass yield by *X. campestris* MO-03 in molasses fermentation media.

As seen in Figure 4, the addition of organic nitrogen sources significantly improved ($p < 0.05$) the quality of the xanthan gum; the pyruvate content of xanthan was 2.85–3.2% ($w/w$) higher than that of the CM (2.1%, $w/w$). The results showed that the maximum pyruvate yield was obtained with CFP (3.2%, $w/w$) and followed by TP (3.05%, $w/w$). Organic nitrogen sources were found to be good nitrogen sources for the production of xanthan gum with higher pyruvate contents [30,34]. As mentioned above, the stimulatory effect of these organic nitrogen sources may be due to the availability of soluble amino acids and minerals in the fermentation broth. It has been shown that the viscosity behavior depends on the pyruvate content of the xanthan gum. The viscosity increases with increasing pyruvate content. The pyruvate content of xanthan gum for food, ophthalmic composition, and oil recovery should be at least 1.5% [34], 2.5% [37], and 0.1–1% [38], respectively. Many of these applications, except oil recovery, need a quality xanthan gum having a pyruvic acid content of about 3% [34].

As seen in Table 1, many researchers have used molasses as a low-cost carbon source for xanthan gum production. The xanthan gum production achieved in this study with *X. campestris* MO-03 (20.5 g/L) using sugar beet molasses and CFP at determined optimal concentrations was much greater than the productions obtained in several previous studies.

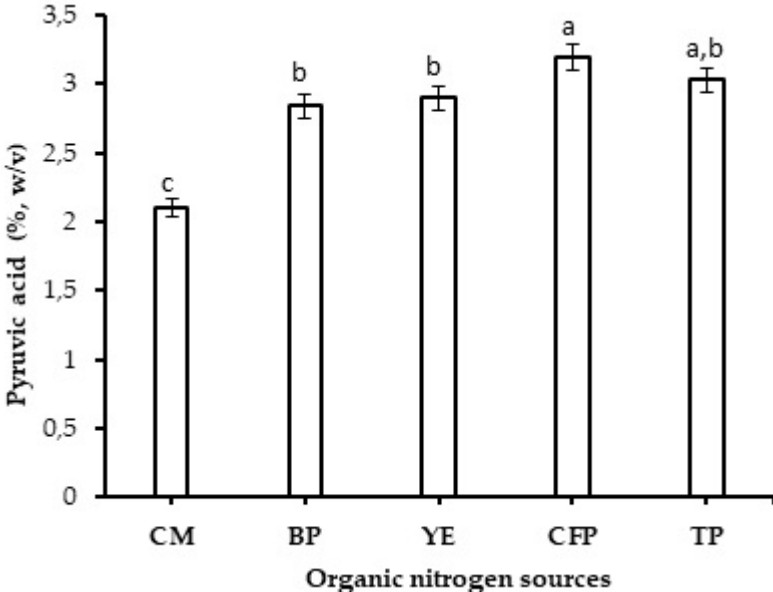

**Figure 4.** Effect of the organic nitrogen sources on the pyruvic acid content in xanthan gum in molasses fermentation media. Values with the same letter are not significant ($p < 0.05$).

**Table 1.** Xanthan gum production in molasses fermentation media.

| Media | Xanthan Gum (g/L) | Pyruvic Acid Content (%) | References |
|---|---|---|---|
| Molasses (17.5%), 4 g/L $K_2HPO_4$, 10 g/L yeast extract | 53 | 1.2–2.3% | [8] |
| Molasses (6%), 10 g/L yeast extract, 3 mL mineral solution | 17.1 | ND | [15] |
| Molasses (10%), corn steep liquor (2%), phosphate (0.1%), citrate (0.1%) | 22.8 | ~2 | [33] |
| Molasses (10%), 10 g/L $KH_2PO_4$, 0.3 g/L $MgCl_2$, 0.5 g/L citric acid | 27.9 | ND | [39] |
| Molasses (10%), 3.0 g/L $K_2HPO_4$, 0.25 g/L $MgSO_4$, 1.4 g/L $(NH_4)_2SO_4$, 1.0 g/L $CaCO_3$ | 19.8 | ND | [40] |
| Molasses (8%), 3.0 g/L $KH_2PO_4$, 0.25 g/L $MgSO_4$, 0.25 g/L $CaCO_3$, 4 g/L CFP | 20.5 | 3.2 | This study |

ND: Not determined.

## 3. Materials and Methods

### 3.1. Microorganism

*X. campestris* strain MO-03 (GenBank: KF939142.1), a wild-type strain, was used throughout this study. This isolate was isolated from the infected leaf of a cabbage [23].

### 3.2. Hydrolysis of Chicken Feathers

All chemicals used in the present study were of analytical grade and supplied from Sigma–Aldrich (St. Louis, MO, USA) and Difco (Detroit, MI, USA). Chicken feathers were obtained from the poultry farm of Zonguldak, Turkey. CFP was produced for this study according to Ozdal and Kurbanoglu [21].

### 3.3. Pretreatment of Molasses

Beet molasses obtained from a local sugar mill. They were diluted with distilled water in order to obtain 60 g/L total sugar concentration. This molasses solution was adjusted to pH 7.0 with 1N $H_2SO_4$ and the liquid was boiled for 5 min. The molasses solution was allowed to stand for 24 h and centrifuged for 10 min at $2500 \times g$. Finally, supernatant of the pretreated molasses was diluted to 40 g/L sugar for xanthan gum production.

### 3.4. Media and Culture Conditions

One loop of cells grown on Tryptic Soy Agar plates for two days were used to inoculate a 250 mL flask containing 50 mL of yeast extract malt (YM) liquid medium containing (g/L) malt extract 3.0,

peptone 5.0, yeast extract 3.0, glucose 20.0 at pH 7.0. The culture was incubated at 30 °C for 48 h in an incubator shaker at 200 rpm. For the xanthan gum production, 5% (*v/v*) of the inoculum (OD$_{600}$ 1.5) was added to 50 mL of the control and production media in a 250 mL flask and incubated in a shaker at 200 rpm, 30 °C for 96 h. The control medium (CM) was composed of sugar beet molasses (total reducing sugar) 40 g/L, KH$_2$PO$_4$ 3 g/L, MgSO$_4$ 0.25 g/L, (NH$_4$)$_2$SO$_4$ 1.4 g/L, and CaCO$_3$ 0.25 g/L. To determine the effects of CFP on xanthan gum production, 1–6 g/L CFP were added to the production medium. CFP was used as the nitrogen source instead of (NH$_4$)$_2$SO$_4$. The pH was adjusted to 7 before autoclaving at 121 °C for 20 min. Later, CFP was compared with three commercial organic nitrogen sources (tryptone (TP), bacto peptone (BP), yeast extract (YE)) at the optimal CFP concentration.

### 3.5. Analytical Methods

At regular intervals (24–96 h) of fermentation, the microbial growth, residual sugar, and xanthan gum were determined. Biomass was determined by centrifuging the fermented broth at 2000× *g* for 10 min and then washed twice with distilled water before following another centrifugation. Finally, the biomass was dried in an oven at 70 °C to constant weight and then weighed. Xanthan gum was obtained by precipitating the cell free fermentation broth with two volumes of 95% ethanol. The solid fraction was washed with ethanol, and then it was dried for 18 h at 90 °C and weighed [30]. Pyruvate content was estimated according to the method of Sloneker and Orentas [36] after hydrolysis of xanthan samples in 0.1 N HCl at 100 °C for 4 h. The phenol sulphuric acid method of Dubois et al. [41] was used for the reducing residual sugar, after biopolymer recovery. All experiments were replicated twice, and averaged values are presented in this study.

### 3.6. Statistical Analysis

All analyses were performed in triplicate. The statistical analyses of the data were carried out one-way analysis of variance (ANOVA) using software package SPSS15.0 (SPSS Inc., Chicago, IL, USA). A $p < 0.05$ was considered statistically important.

## 4. Conclusions

As is known, agricultural and industrial wastes continue to increase due to world population increase, thus these wastes pose a big problem. Sugar beet molasses and chicken feathers are cheap, abundant, and easily accessible waste. Therefore, the usability of CFP as both nitrogen and mineral salt source in the presence of sugar beet molasses for the production of xanthan gum was determined. Among organic nitrogen sources, maximum xanthan gum production was obtained with CFP. The use of CFP increased the pyruvic acid content of xanthan gum. To meet the demand of xanthan gum requirement due to many industrial applications, a low-cost medium is required for the fermentation. Finally, xanthan gum production using sugar beet molasses and CFP as the substrates by the local isolate of *X. campestris* is possible, and a cost reduction may be achieved.

**Author Contributions:** M.O. and E.B.K. carried out the experimental work; M.O. was responsible for the writing and E.B.K. contributed to the discussion during the research work.

**Funding:** This research was funded by the Atatürk University Research Fund as a research project (Project No: 11-116).

**Acknowledgments:** This research was supported by a grant from the research funds appropriated to Ataturk University, Erzurum, Turkey.

**Conflicts of Interest:** The authors declare no conflict of interest.

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
