# Peer review of "Use of Chicken Feather Peptone and Sugar Beet Molasses as Low Cost Substrates for Xanthan Production by Xanthomonas campestris MO-03"

_fermentation, doi:10.3390/fermentation5010009_

Reviewer 1 Report

The manuscript is  good prepared and the experiment design is proper. However, the section  3 needs to be reconstruated: Figures and Table should appear, were they are mentioned in the text. I suggest to add some new subsections (for example:

3.1 The effect of organic nitrogen sources

3.2 Pyruvic acid content of xanthan gum

3.3 Xanthan gum production in molasses fermentation media) instead of "3.2. Figures, Tables and Schemes"

Author Response

Dear Reviewer, 

We are thankful to you for your kind suggestions and critical comments. They were essential for the improvement of the manuscript.  We look forward to your kind opinions. 

We would like to thank the reviewer for careful and thorough reading of this manuscript and for the thoughtful comments and constructive suggestions, which help to improve the quality of this manuscript. Based on these comments and suggestions, we have made modifications and careful check of the manuscript.

We look forward to your kind opinions. Below are the itemized responses. 

Reviewer report: 

The manuscript is good prepared and the experiment design is proper. However, the section 3 needs to be reconstruated: Figures and Table should appear, were they are mentioned in the text. I suggest to add some new subsections (for example:

 3.1 The effect of organic nitrogen sources

 3.2 Pyruvic acid content of xanthan gum

 3.3 Xanthan gum production in molasses fermentation media) instead of "3.2. Figures, Tables and Schemes"

 Ans: They are changed as suggested and corrected in the manuscript.

 Two subheadings added. 3.1. Effect of CFP on xanthan gum production

                                    3.2. Effect of organic nitrogen sources on production of xanthan

Reviewer 2 Report

Dear Authors,

The manusctipt entitled “Use of chicken feather peptone and sugar beet molasses as low cost substrates for xanthan production by Xanthomonas campestris MO-03” by M. Ozdal and E B Kurbanoglu explored the effective utilization of sugar beet molasses and checken feather peptone (CFP) for the production of xanthan gum by microbial fermentation of Xanthomonas campestris.  

In this study, the authors demonstrated that the high yield production of xanthan gum containing pyruvic acid was achieved by using sugar beet molasses and CFP at determined optimal conditions.  It is greater than the production levels in several previous studies. These outcomes may contribute to development of the xanthan gum applications for many purposes.  However, there are some major and minor comments on this manuscript. Therefore, it is needed minor revisions and explanations for acceptance to the journal of Fermentation.

> Major comments:

As described in the Introduction, sugar beet molasses also contains a lot of trace elements such as minerals and vitamins.  For example, amino acid fermentation by Corynebacterium glutamicum is critically affected by biotin in the sugar cakane molasses.  In this study, did the authors confirm the effects of these trace elements for the production of xanthan gum?

As for the biomass production, I agree with that the final biomass production at 96 h depends on the CFP supplement concentration in the medium.  However, at the early phase of fermentation, it was observed that high concentration of CFP apparently inhibits the biomass production.  The authors should explain about this phenomenon.

What is molecular weight distribution of the produced xanthan gum polysaccharide in this study?

The authors insist on  that the xanthan gum production is dependent on the CFP concentration in media.  However, it seems to be not exact. At least, in Figure 1c, the concentration dependence of CFP on the xanthan gum productivity was only seen up to 4 g/L of CFP concentration.  It is not mentioned about why its phenomenon occurred.  In addition, in Figure 2, the changes in pyruvic acid content of xanthan gum are out of sync with the changes in xanthan gum production in Figure 1c. What are you explain it?

> Minor comments:

line 36: The production of xanthan gum >> The production of xanthan gum by microbial fermentation

Line 77:  at 3500 rpm  please convert into centrifugal acceleration unit (xg).

Line 99: xanthan samples in 0.1 M HCl >>> xanthan samples in 0.1 N HCl

line 93:  both at 10,000 rpm  convert into centrifugal acceleration unit (x g).

line 114: xanthan gum and bacterial growth >>> xanthan gum production and bacterial growth

line 121: However, the xanthan concentration >>> However, the xanthan gum concentration

line 116-117:  The authors claim that the xanthan gum production was observed to be dependent on the CFP concentration.  However, I do not think so at least.  Because in Figure 1b and 1c,  

Line 120: ...concentration in the CM + 4 g/L CFP was observed at 60 h (Figure 1c).

line 124:  Remove “According to Figure 1b,”.

line 116:   The description “ The xanthan gum concentration was observed to be dependent on the CFP concentration.” is in conflict with the described later “line 126-127:  These results suggested that the increase in xanthan gum concentration was not dependent on the increase in the growth related.”.  I think it misleads the readers. Please change the representation of either sentence.

Line 156 & 173:  I think that P>0.05 is typos of P<0.05.  If it is not true, what is your null hypothesis for the effect of CFP supplementation?  There are same typoes in other parts of this manuscript. Please confirm them.

Line 164: I could not find the maximum biomass yield TP (4.88 g/L) in Figure 3.  Please confirm data and revise to exact data.

Line 173-176:  It seems that the data described in the text (line 173-176) are inconsistent with the values can be implied from Figure 4. Please confirm the data and the original Figure.

Line 218-219:   “chicken feather peptone” should be corrected to “CFP”.

Line 219:  X. campestris should be changed italic style.

Line 231:  remove “()”

line 234-235:  This article is opened as electric journal, so it should be described as “Doi: 10.1002/APP.42035”.

Line 259 & 277:  What is meaning of “3 Biotech” in the journal title?

Line 261: remove “(2018)” and should be unify the article title.

Line 267:  remove “(2014)”

line 272:  “(12.10.1018) should be corrected to “(12.10.2018)”.

line 273-275:  remove “(2016)” and should be unify the article title. In addition, “180: 1401-1415.” should be corrected to “180, 1401-1415.”.

line 278:  remove “(2014)”

line 282-283:  The article title characters should be corrected.

Line 283:  The year “2018” should be changed to bold type.

Line 289:  The year “2018” should be changed to bold type.

Line 291:  The journal title “Process Biochem.” should be changed to italic type.

Line 298:  The journal title “J. Sci. Food Agric.” should be changed to italic type.

Line 301-302:  The year “2011” should be changed to roman type and the volume “51” to italic type.

Line 307-308:  The article title should be unify.

Line 328:  remove “(2013)”

line 331:  The article title should be unify.

In the title of this manuscript, the strain number of Xanthomonas campestris is presented as MO-03, but it is as MO-3 in the text. Please unify the term.

In this manuscript, especially in Figures and their legends, there are many notation variability of terms (e.g. pyruvic acid / pyruvate, xanthan / xanthan gum).  Please unigy their terms.

In Figure 1, the legends pertaining to the drawing should be represented in the explanation. In Figure 1c, Y-axis title “Xanthan” should be changed to “Xanthan gum”.

In the title of Figure 1, the order of (b) and (c) is an inverse.

In the Figrue 2, the title of Y-axis “Pyruvate” should be corrected to “Pyruvic acid”.

In the title of Figure 2, “.... are not significantly different at the p ≤  0.05 level.” should be corrected to “... are not significantly different at p ≥0.05.”.

In Figure 3, why do not data regarding the xanthan gum and biomass productions in the control CM medium?  In addition, in the Figure 3, the order of column is unsuitable for this study.

It seems that the title of Figure 3 should be improved as “Effect of organic nitrogen sources on xanthan gum production and biomass yield by X. campestrisMO-03”.

The title of Figure 4 shoud be corrected to “ Effect of the organic nitrogen sources on the pyruvic acid content in xanthan gum.” to unify with the title of Y-axis.
